# ElastoMeric Infusion Pumps for Hospital AntibioTICs (EMPHATIC): A Feasibility Study

**DOI:** 10.3390/antibiotics14111122

**Published:** 2025-11-07

**Authors:** Joseph J. Spencer-Jones, Stuart E. Bond, Nicola Walker, Jade Lee-Milner, Julie Thompson, Damilola Mustapha, Annam Sadiq, Achyut Guleri, Jayanta B. Sarma, Liz Breen, Jonathan A. T. Sandoe

**Affiliations:** 1Mid Yorkshire Teaching NHS Trust, Wakefield WF1 4DG, UK; stuart.bond@nhs.net (S.E.B.); nicola.walker41@nhs.net (N.W.); jade.lee-milner@nhs.net (J.L.-M.); damilola.mustapha@nhs.net (D.M.); annam.sadiq@nhs.net (A.S.); achyutguleri@gmail.com (A.G.); jayanta.sarma@nhs.net (J.B.S.); 2National Institute of Healthcare Research, Leeds LS1 2TW, UK; 3School of Medicine, Faculty of Medicine and Health, The University of Leeds, Worsley Building, Leeds LS2 9JT, UK; j.sandoe@leeds.ac.uk; 4Department of Pharmacy, School of Applied Sciences, University of Huddersfield, Huddersfield HD1 3DH, UK; 5School of Pharmacy & Medicinal Sciences, Faculty of Life Sciences, University of Bradford, Bradford BD7 1DP, UK; l.breen@bradford.ac.uk; 6Leeds Teaching Hospitals NHS Trust, Leeds LS1 3EX, UK

**Keywords:** antibiotics, elastomeric, beta-lactams, intravenous, inpatient, flucloxacillin, benzylpenicillin, piperacillin/tazobactam, antibiotic therapy

## Abstract

Background: Elastomeric infusion pumps (EMPs) are safe and effective for administering outpatient intravenous (IV) antibiotics. We hypothesized that EMPs may provide benefits in the inpatient setting. This study aimed to assess the feasibility of giving IV antibiotics using EMPs to adult inpatients and to identify barriers and facilitators for their implementation. Methods and Objectives: Patients who were 18 years of age and over requiring at least seven days of IV flucloxacillin, benzylpenicillin or piperacillin/tazobactam and who were clinically stable were eligible. We collected quantitative data for feasibility, clinical outcomes and intervention acceptability. We applied an implementation research framework to help triangulate the data. Analyses were descriptive, with the intent of preparing for future studies. Results: IV antibiotics from 94 EMPs were administered to nine patients, with five patients completing treatment with an EMP. Five of the six patients surveyed said they would use EMPs again. Nurses felt EMPs were safer, less time consuming and improved working conditions. IV antibiotics via EMPs cost GBP 32.50 (GBP 3.35–GBP 83.44) more per day than intermittent infusions. Residual volume in EMPs was an issue which resulted in reduced antibiotic doses being delivered. The main facilitators to use of EMPs in the inpatient setting were adaptability, tension for change, recipient centeredness and needs of the deliverers. The barriers were lack of advantage, critical incidents and cost. Conclusion: This proof of concept feasibility study shows that it may be feasible to use EMPs in the inpatient setting. There is potential to improve patient and staff experience; however, cost and residual volume are potential barriers to implementation, with further studies required.

## 1. Introduction

Antimicrobial resistance (AMR) is a major threat to health worldwide [1]. The need to optimize antibiotic use is vital, considering that an estimated 1.27 million deaths globally were attributed to AMR in 2019 [2]. In the United Kingdom one in five hospitalized patients in the UK are prescribed an IV antibiotic [3]. Preparation and administration of each antibiotic infusion by a nurse takes around 22 min [4]. However, with a shortage of 40,000 nurses in the NHS [5], 61% of nurses report that they are “too busy to provide the level of care they would like” [6]. This contributes to potential errors, including missed antibiotic doses [7], which can help to drive AMR [8] as well as worse outcomes.

Elastomeric infusion pumps (EMPs) are used successfully to deliver intravenous analgesia, chemotherapy and cardiology treatments [9,10], though mainly in the outpatient setting. EMPs are recommended for administration of outpatient IV antibiotics via outpatient parenteral antibiotic therapy (OPAT) services [11], where this approach is safe and effective [12]. Antibiotic delivery using EMPs has several advantages, including the following: optimized administration of time-dependent antibiotics, improved patient freedom [13], less administration time than intermittent doses and the possible reduction of the burden of AMR by enabling the administration of the narrowest spectrum antibiotic possible, optimizing serum levels and reducing the number of missed doses. Although studies have been inconclusive regarding mortality, continuous infusions of time-dependent beta-lactam antibiotics have been shown to improve clinical cure and target plasma concentrations [14,15,16].

Despite the benefits that EMPs provide, a recent systematic review found limited evidence of their use to deliver antibiotics in the inpatient setting [17]. We aimed to test the feasibility of giving intravenous antibiotics to adults via EMPs in an inpatient setting and to better understand the barriers and facilitators for their implementation in this setting [18].

## 2. Results

In total, 243 patients were screened for eligibility. Twelve patients were deemed suitable and were approached for enrolment onto the study; nine patients consented to take part in the study (Figure 1). Table 1 shows the characteristics of those who consented to take part in the study. Flucloxacillin was the most frequently used antibiotic (n = 5), followed by benzylpenicillin (n = 3) and piperacillin/tazobactam (n = 1). Indications were infective endocarditis (n = 5), discitis (n = 3), osteomyelitis (n = 1) and prosthetic joint infection (n = 1); further information about the pathogens that were treated is available in Appendix A, Table A1. In total, 94 EMPs were administered over the course of the study.

### 2.1. Primary Outcome—Feasibility

A total of 4.9% (12/243) of all patients screened were identified as being eligible for enrolment onto the study (Figure 1). Median duration of EMP treatment was 5 days (1–29 days). Median duration of all antibiotic treatment (initial therapy plus EMP) was 16 days (7–50 days, Table A1). Of the nine participants who received at least one EMP, five (56%) completed the planned course of treatment entirely with an EMP (Table A1, Figure 2). Two had legitimate clinical reasons for requiring EMP use to stop. Two other participants withdrew, one due to the dislike of the vascular access device and another had concerns with residual volume and inability to take part in physiotherapy sessions.

### 2.2. Secondary Outcomes

#### 2.2.1. Clinical Cure

Clinical cure was achieved in six patients (Figure 1, Table A1). Of the five patients who completed full treatment with the EMP, three had clinical cure. No participants died within 30 days of diagnosis, although three participants died within 90 days of diagnosis, two of whom completed the full treatment with EMPs.

#### 2.2.2. Adverse Events

All EMPs had a total volume of 240 mL, with high residual volume defined as >36 mL reaming after 24 h. This was seen in 68% (54/81) of the EMPs measured. The average residual volume of the 54 EMPs that had over 36 mL remaining was 61 mL (range 39–200 mL). Of the Vygon (UK) Ltd. (Swindon, UK) Accufuser^®^ EMPs that were measured, 64% (37/58) had a residual volume over 36 mL. Of the 23 Baxter LV10^®^ EMPs that were measured, 70% (16/23) had a residual volume over 36 mL. In addition, four pumps had a clamp left on, two EMPs were stopped because additional IV treatment was required and two had administration times changed.

Two serious adverse events occurred during the study (Table A1). One patient developed deranged liver function tests with the decision made to switch to an alternative treatment, therefore stopping the EMP. The second patient developed a PICC line infection after completion of treatment. Both patients died within 90 days of diagnosis. In addition, one blocked PICC line was reported, which was unblocked using urokinase as per the trust policy.

#### 2.2.3. Procurement Process

A total of 129 EMPs were ordered. Data are available for the full process of ordering five batches for delivery, totaling 56 EMPs. Median time from prescribing to delivery to the ward was 4.5 days (108 h, range 50–141 h). The median time from placing an order to receiving a delivery was almost 2 days (45 h, range 33–166 h). Quality control (QC) took 30 min for each batch; however, due to the opening hours of the department, the median time for assessment and release was 2.4 h (0.25–18 h). Twenty-six observations were made when dispensing the EMPs; the median dispensing time, i.e., the time it took for a prescription to be validated by a pharmacist and then dispensed, was 32 min (0.58–3.3 h).

#### 2.2.4. Administration

In total, 94 EMPs were administered and 23 EMP administrations were timed, with the median time to set up and start the running of an EMP being 11 min (5–20 min).

There were delays in the administration of 74/94 EMPs (78%) (median 20 min; range 2 min–5 h). No doses given via EMP were missed, compared to 20/115 (17%) intermittent doses missed and 8/115 (7%) intermittent doses delayed, across all nine patients. There were four errors, with clamps remaining in place after an EMP was connected. The total number of wasted pumps was 34, with 1 pump unaccounted for. The reasons for wastage included change in treatment (n = 19), withdrawal from study (n = 5), adverse effects (n = 4, see Section 2.2.2), expiration (n = 3) and incorrect storage (n = 3).

#### 2.2.5. Costs Analysis

A basic cost analysis was undertaken (shown in Table 2). A breakdown of workings can be found in the Appendix A, Table A2, Table A3, Table A4, Table A5 and Table A6. Average costs were used with ranges due to the price differences between the different antibiotics and doses used. Costs also included nursing time based on the information above for EMP and intermittent administration. A delivery charge was added for each batch of EMPs ordered. This totaled a further GBP 2073.36.

#### 2.2.6. User Acceptability

Of the nine patients in the study, six provided feedback on user acceptability. Figure 2 shows that patients receiving EMPs were satisfied when it came to daily activities, seeing improvements in getting dressed, mobility around the ward and quality of sleep. Five of six (83%) surveyed patients said they would receive treatment with EMPs again if given the option.

Of the 42 nurses who administered an EMP, 9 nurses completed the survey (20% response rate). Eight were satisfied with the training and SOP provided prior to starting an EMP on their ward. All nursing staff felt that the EMPs were safer for staff and patients, were less time consuming than intermittent dosing and that routine use of EMPs would improve working conditions (Figure 3).

#### 2.2.7. Consolidated Framework for Implementation Research Analysis [19]

##### Innovation Characteristics

Adaptability was the main facilitator in this domain, as EMPs were easily refined from the outpatient to the inpatient setting. There was one report of EMPs preventing other therapies from taking place “the line from the pump was getting caught during transfer or doing physio sessions and this was very painful” (Patient 5).

As this was an internally developed innovation, it was well received by nursing and medical teams.

Despite nurses reporting positive responses regarding workload, safety and patient experience, the relative advantage of the innovation was impacted by the residual volume seen across multiple EMPs. This contributed to one patient withdrawing from the study due to “concerns about residual volume as well as inability to take part in physiotherapy sessions” (Patient 5).

The complexity of the intervention was also a barrier; while most of the residual volume was unexplained, some issues were complexity-related such as “nurse noticed that pump was not emptying as line was clamped” and “nurse contacted pharmacy to ask how to connect a pump”, while one nurse responded “could do with proper training as a team and have chance to ask questions” (Nurse 8).

Though not reported in the surveys, the cost analysis showed that EMPs cost more per day than standard therapy.

##### Outer Setting—This Is the Setting in Which the Inner Setting Is Based: In This Case National and Local NHS Health Systems

Patient safety, improving working conditions and reducing workload were all key areas contributing to the policies and law domain. Despite improvements in workload, reports of being “unable to check pump sometimes due to ward workload” (Nurse 5) and “delay to removal from fridge due to workload” (Nurse 6) were still reported.

The amount of residual volume and the lack of other hospitals using EMPs in this setting is a barrier as there is a lack of market pressure, both at the local and national levels, to drive implementation.

##### Inner Domain—This Is the Setting Were the Innovation Takes Place: In This Case the Individual NHS Trust/Wards

Facilitators of adoption of EMPs—The improvement of staff working conditions and patient experience, e.g., “patients can get some sleep at night instead of waking them up to administer” (Nurse 8), while reducing workload were strong identifiers that this innovation was both patient and nurse focused. Nurse responses provided insights on the tension regarding change, particularly around working pressures and patient safety with “less chances of missing doses” (Nurse 8).

The innovation was compatible with existing systems and work processes. Nurses reported that SOPs, training and advice was useful; “graphic charts help a lot” (Nurse 7) and were easily accessible.

Barriers to the adoption of EMPs—Workforce infrastructure was a barrier in relation to insertion of vascular access (PICC lines). At the time of the study, no vascular access service was available at the trust. There were two delays to treatment due to a lack of PICC line access. Quality control and assurance processes were also problematic. While most pumps were delivered within 48 h, it took more than double the delivery time for these to be processed and delivered to the ward once they were on site.

##### Project Characteristics

Facilitators of adoption of EMPs—Innovation deliverers (nurses) reported that using the EMPs reduced workload pressures and improved working conditions and patient safety. Innovation receivers (patients) favored pumps when it came to completing personal care and would use EMPs again. Regarding capability, the EMPs used in this study were pre-filled, with nurses required to complete their usual checks, perform PICC line maintenance and then attach the EMP to the line. Nurses already had training on administering IV therapy and most on PICC line maintenance, making the innovation as simple as possible.

##### Implementation Process

Facilitators of adoption of EMPs—This involved understanding of the benefits of EMPs realized in OPAT and how these might meet the needs of the deliverers (nurses) and receivers (patients). Often, hospital nurses and patients were not aware that EMPs are used in the community to good effect; therefore, explaining the benefits they provide and the potential for this to be replicated in the inpatient setting helped. This was especially evident when training nurses.

Barriers to the adoption of EMPs—Engaging with nurses was challenging due to work pressures, so not all nurses received training. One suggestion proposed was “[they] could do with proper training as a team and have chance to ask questions” (Nurse 8). Only 20% of nurses actually completed the survey, so engagement with feedback was low.

## 3. Discussion

This study confirmed the feasibility of providing IV antibiotics to adult inpatients using EMPs; the eligibility criteria was met, while potential barriers and facilitators that may affect future research and wider implementation were identified.

The acceptability of the idea of using EMPs for administering IV antibiotics to patients was confirmed by the high proportion of eligible patients who consented to take part (75%). However, the conversion rate of those screened (n = 243) for being eligible was low (5%), with a short course length being the main reason for ineligibility. Research shows the benefits of oral route delivery over IV therapy, with increasing evidence for oral antibiotics in being capable of combatting complex infections and thereby reducing the need for administration of long IV courses [20,21]. This, combined with the logistics around ordering and delivery of EMPs, resulted in limiting the inclusion criteria to those who would require over 7 days of IV therapy. This may be improved by using different methods to fill the EMPs. For example, nurses could fill EMPs at the patient’s bedside. This would reduce ordering, delivery and quality control time. Switching from pre-filled to nurse-filled EMPs in the OPAT setting has been shown to reduce costs, while also not significantly affecting visit times [22,23]. Both studies reported improved emptying with the nurse-filled EMPs, with no reports of line infections or complications in Leeds, despite pumps being prepared in a non-sterile environment [22,23]. This method would also remove the need for QC/QA assessment, as the EMP is produced on site and fresh. Pre-filled EMPs are classed as unlicensed medicines and therefore are required to be assessed on delivery. Improving this process and being satisfied of the quality of the product may help to alleviate this in the future. Understanding how pre-filled and nurse-filled pumps affect implementation will be vital for further studies and is one area we will focus on.

Improving ordering and delivery, while also using a more effective means for identifying potentially suitable patients, is needed. As a result of the requirement for long planned courses, all the participants had complex infections with a known high mortality; 30-day all-cause mortality of infective endocarditis, for example, has been reported to be around 11% [24,25] and 14% for vertebral osteomyelitis [26]. This likely contributed to the mortality rate seen in our study.

There were some clear advantages to EMPs. On average, only 22 min is required to prepare and administer intermittent IV antibiotics [4]. EMPs, however, only took 11 min on average (5–20 min). Had EMPs not been used, there would have been 485 intermittent doses required to be administered to patients. EMPs were estimated to save 160 h of nursing time for the nine patients included in the pilot. The costs for nursing administration time with EMPs were estimated to be 10 times less than for intermittent dosing. No missed doses were observed for EMPs, though delays were seen, compared with 17% for intermittent dosing, a similar rate to that reported in other studies [27,28]. A National Patient Safety Alert (NPSA) in February 2010 highlighted the importance of avoiding missed doses when they reported 27 deaths, 68 severe harms and 21,383 incidents due to delayed or omitted doses [29]; antimicrobials were involved in 9 deaths and 22 severe harm cases [29]. EMPs were favored by participants and staff for ease of use (relative to intermittent infusion) and perceptions of safety and improved patient experience.

The acquisition of EMPs through the trust ordering system was slow, taking over four days for delivery in a setting where EMPs had to be ordered-in rather than prepared on site. Although often short, there were delays in administering the EMPs to most of the participants in this feasibility study, highlighting the need for more robust systems of delivery to wards and training. One participant was unable to carry out physiotherapy due to the EMP being connected and withdrew from the study. As EMPs do not require a power source or drip stand, we expected that this may improve patient mobility as seen in other studies [30], which was borne out by the patient survey, so the inability to participate in physiotherapy was unexpected and needs further study to understand if or how this can be overcome.

A residual volume of over 36 mL was seen in over half of the EMPs and was also a reason for withdrawal from the study (Patient 5). The devices in this study were mainly Vygon Accufuser^®^ (Vygon UK Ltd., Swindon, UK), being used in 67 out of 94 EMPs. Baxter LV10 Infusor^®^ (Baxter Healthcare Corporation, Chicago, IL, USA) devices were used for 27 of 94 EMPs. Rates of residual volume were similar: 65% for Vygon and 70% for Baxter. These devices were reported as defective products to the Medicines and Healthcare Regulatory Agency (MHRA) and comprised a critical incident that was not expected during the study, requiring further investigation. However, this study could not determine whether the reduction in total 24-hourly dose delivered to the patient affected the time above the minimum inhibitory concentration (MIC), which is thought to be vital for optimal penicillin dosing [16,31], though no treatment failure was reported. Exploring this in future work, using MIC and plasma blood monitoring, would help to understand the effects of residual volume on patient care and the impact that this could have with antimicrobial resistance.

A previous study comparing different types of EMPs for chemotherapy found that the Accufuser® had the second largest average residual volume of 2.81 mL [32]. Another reported that 40% (n = 50) patients had residual volume with an EMP and seven of these had a volume that exceeded 20 mL [9]. One retrospective study looking at EMPs malfunction, injury and death found 384,285 reports made to the US Food and Drug Administration’s Manufacture and User Facility Device Experience (MAUDE) body between 2010–2018 [33]. This study focused specifically on postoperative analgesia, with 4093 reports included, of which 88.5% were classed as malfunctions of the EMP, with 0.2% of cases resulting in death [33]. Residual volume is affected by several factors, including temperature [34]. A raised temperature can lead to a lower viscosity and therefore the quicker emptying of the EMP. While a low temperature can result in increased viscosity and a slower emptying of the EMP [13]. Temperature can also affect the stability of the antibiotic [13], resulting in degradation of the antibiotic or precipitation out of the solution. The latter was seen with acyclovir; when exposed to in-use temperatures of 32 °C, a massive precipitate formed [35], which could potentially block the EMP or line, resulting in delayed or incomplete emptying. Most studies confirming the stability of antimicrobials in elastomeric devices have been undertaken in standardized laboratory conditions, with examples in real-life conditions lacking [36,37]. However, recent studies, as well as stability data, suggest that all three antibiotics used in this pilot are stable if stored and used correctly [38]. Furthermore, a recent systematic review found that high quality data to support stability of antimicrobials in EMPs is now available [39].

While EMPs bring advantages such as simplicity and a lack of power source, issues with residual volume and flow rate present patient safety concerns that could contribute to treatment failure. Control of flow rate with EMPs is imprecise compared with modern electronic devices, and a study currently in the recruitment phase is comparing nurse and patient satisfaction with EMPs compared with portable electronic devices [40]. Secondary outcomes include cost analysis, pump-related complications, temperature variations and actual antibiotic dose delivered [40].

Our study observed much larger residual volumes than previously described [9,32], which requires further investigation. The impact of this on clinical outcomes cannot be assessed from this feasibility study but highlights the importance of this issue in future work. There will be a requirement for better staff and patient education around their proper use. Future work may also need to consider the psychological and clinical effects this has on patients, especially given the withdrawal of one patient due to concerns of inadequate dosing.

Other adverse events reported were PICC line related (infection and blockage), which is a recognized problem with all types of infusions and not a specific problem to EMPs.

The estimated daily cost of EMPs including drug and ancillary cost varied depending on the antibiotic and dose used. Without delivery fees, the average difference in daily cost for EMPs versus intermittent infusions ranged from GBP −18.71 to GBP 61.38. This range was seen due to 12 g of flucloxacillin per day and 7.2 g of benzylpenicillin per day requiring doses to be given six times per day, costing GBP 40.02 in nursing time. EMPs saved, on average, GBP 28.38 in nursing time per day of EMP compared to average intermittent costs. However, given the extra costs of the pumps, this was only offset when dosing six times a day is required. When delivery charges are added, the costs per day using an EMP compared to intermittent favors intermittent dosing. EMPs cost an average of GBP 32.50 (GBP 3.35–GBP 83.44) extra per day. Scaling this up would present a cost pressure, but through alternative methods of sourcing and delivery it may be possible to reduce costs, as well as potentially only focusing on those antibiotics that need to be given more than four times per day. A full health economic analysis was beyond the scope of this study and further work is required to fully assess the cost effectiveness of EMPs for inpatients. While cost is a key consideration, patients and staff experience are paramount. The majority saw improvements in daily activities, sleep quality and mobility. Nursing staff felt EMPs were better compared to intermittent infusions for both staff and patients, consistent with findings in OPAT studies [12,41]. The majority agreed strongly that EMPs would improve working conditions, something that is key given the current climate of the NHS.

Implementing change in healthcare organizations is difficult, with failure rates between 30–90% [42]. Even when interventions are safe, effective and evidence based, this does not always translate into adoption in practice [43]. As with other studies, the Consolidated Framework of Implementation Research was used to analyze the data obtained rather than as a data collection tool [44]. The main facilitators seen in the Consolidated Framework of Implementation Research were adaptability, tension for change, recipient centeredness and needs of the deliverers. The intervention (use of the EMP) was well received by nursing staff due to the impact on improving working conditions, a key driver for change, especially given staffing shortages within the NHS [45]. It also aligned with the Trust’s mission statement of “providing high-quality health care services”. Further work is needed to understand what this improvement means for nurses, patients and the NHS. The adaptability of the innovation to the inpatient setting was a facilitator, as only minor changes were required compared to the outpatient process. There is potential to further adapt the model of ordering, moving from named patient to batch ordering, but this may require policy changes both locally and nationally. Recent changes in policy may also be key to implementation with a move towards ready-to-administer injectable medication. This is due to the positive impact of ready-to-administer injectables during the COVID-19 pandemic, with the preference of nursing staff for using these [46] and the move to treat more people at home. This may help drive the use of EMPs in this inpatient setting.

Lack of ‘relative advantage’, whereby the intervention does not actually provide the benefit it set out to, in this case due to large amount of residual volume was a key barrier. Critical incidents related to reduced dosing due to residual volume and higher cost were the other main barriers to this innovation. More research is required to assess the impact of this on patient outcomes as well as consumption and AMR, but this was beyond the scope of the study. Despite the EMPs being pre-filled, reducing the need to reconstitute antibiotics and therefore reducing complexity of administering the device, further steps were added to the system that introduced more complexity; this was especially the case with the required extra vascular access and the ordering and receipt of the EMPs, all of which were required to be coordinated at the same time to reduce waste. Complexity was also an issue when using the device, with occasions where clamps were left in place resulting in no dose being administered.

## 4. Limitations

As this was a small single-center study undertaken in a district hospital, the results are only hypothesis generating and give a direction for future work. The realized barriers and facilitators are contextual and may not be the same in other centers.

Given the delay with delivery and the need to be assessed by quality control, only those patients who required over 7 days of treatment were considered. Other methods of procurement and preparation, such as nurse-filled EMPs, were not considered in this study, but if explored this may allow a change to the eligibility criteria.

Patients and staff feedback was obtained from one-time-only surveys, which may affect the data given; additionally, feedback on daily activities was self-reported.

Every effort was made to fully record the process from ordering to administration, but practically this was not possible. The study team’s working hours and clinical commitments resulted in this being performed for just five batches. Several EMPs were wasted due to withdrawal from the study; these had to be destroyed, which is potentially another additional cost of the EMPs compared to intermittent doses that were not included in the cost analysis. The authors note the lack of randomization as a limitation; however, this feasibility study will help to aid the design of future studies to address this.

## 5. Materials and Methods

### 5.1. Primary Objective

The primary objective was to assess the feasibility of administering IV antibiotics to adult inpatients using EMPs, while the secondary objectives concerned understanding the processes involved for further clinical evaluation and implementation. This included identification of any barriers and facilitators.

### 5.2. Secondary Objectives

(a)To describe the antibiotic pathway for each patient;(b)To measure early termination of the planned period of IV antibiotics (including reason);(c)To evaluate user acceptability (medical, pharmacy, nursing) using a survey;(d)To measure antibiotic prescribing costs, including costs of ancillary items (IV giving sets, wipes, flushes, diluents);(e)To measure residual volume remaining in the pumps at the end of the infusion;(f)To measure the ordering and procurement process parameters;(g)To measure serious adverse events, including *Clostridioides difficile* diarrhea and death;(h)To measure intravascular access complications, including infection, thrombosis and line blockage;(i)To identify the barriers and facilitators to implementation.

A single-center, open-label, prospective feasibility study was conducted at Pinderfields Hospital, an 800-bed district hospital part of Mid Yorkshire Teaching NHS Trust, Wakefield, UK (MYTT), from May 2022 to December 2022.

Admitted patients aged ≥18 years requiring IV flucloxacillin, piperacillin/tazobactam or benzylpenicillin were eligible. Only those requiring 7 days of IV treatment or longer were considered due to the length of time required for the ordering process, delivery time, staff training and IV access requests. Patients were required to provide written or witnessed consent and deemed clinically stable and improving by the consultant in charge. Patients who were unable to provide consent, did not have appropriate vascular access such as a peripherally inserted central catheter (PICC), or had a penicillin allergy and were unsuitable for a penicillin challenge were excluded. Patients on combination IV antibiotics requiring multiple daily administrations, awaiting hospital transfer or who were unsuitable for OPAT were also excluded.

### 5.3. Recruitment

Potential participants were identified from Monday to Friday using the hospital electronic prescribing and medicines administration (EPMA) system (Medchart version 10.1.21.5; Dedalus^®^, Milan, Italy). Microbiology reporting systems iLab (AIX version 7, Agilent Technologies, Santa Clara, CA, USA) and ICNET (Version NG1.9.1.1, Baxter Healthcare Corporation, Chicago, IL, USA) were also used to identify patients with positive microbiology, such as *Staphylococcus aureus* bacteremia.

Patient screening was completed by the antimicrobial stewardship (AMS) pharmacy team and microbiologists. The medical team was approached to confirm patients were clinically stable and improving. The patient was then approached and provided with verbal and written study information. They were then given a minimum of 24 h to consider participation and discuss any questions with a member of the study team. Written consent was obtained and documented in the medical notes.

### 5.4. Measures and Data Collection

Key sources of quantitative data were collected and triangulated as outlined below. Qualitative insights were provided via open-text questions and used to corroborate the descriptive analysis of quantitative data.

The primary outcome was the percentage of consented patients, who completed the planned course of IV antibiotics via an EMPs.

Secondary outcomes included the following:Demographic and clinical data were extracted from the hospitals’ electronic medical records (PPM+, version RC22.04.01—RC22.12.12, Leeds Teaching Hospitals, Leeds, UK) for each patient at baseline.A literature-informed patient feedback survey (see additional file 1) was developed by the study team that included co-author JT, a patient and a public involvement representative at the Trust [40]. Each patient was asked to complete the survey toward the end of their planned course of antibiotics via the EMP. The nursing feedback survey (see additional file 2) contained questions relating to training, information, administration, issues and overall experience.Intermittent dosing and EMP costs (inc. VAT) were calculated and compared, including giving sets, wipes, flushes, diluents and delivery. Due to commercial sensitivity, only the difference in cost between intermittent dosing and EMPs per day was reported. The costs used were 2022/23 prices. Staff costs were estimated based on costings by NHS England for nursing staff of GBP 18.19 per hour [47].Residual fluid volume in the EMPs was measured manually by removing the remaining solution, with a margin of error provided by Lloyds Pharmacy Clinical Homecare (LPCH, Coventry, UK) of 15% (36 mL) of a 240 mL EMP.Number of patients screened compared with those recruited.Clinical cure, defined as the resolution of symptoms and completion of antimicrobial therapy, which was assessed by the hospital medical consultant.Mortality at 30 and 90 days.Reliability of procurement, including time taken for delivery and any delays.Missed or delayed doses.Nursing time to administer.Requirement of intravascular devices.Difference in cost between EMPs and standard treatment (including ancillary items such as IV giving sets, wipes, flushes and diluents).Adverse events related to antibiotics, including blood dyscrasias, serum sickness, allergies, Clostridioides difficile diarrhea and death.Line-related adverse events including blockage, thrombosis and infection.

### 5.5. Data Analysis

We used a basic descriptive analysis for quantitative data. This was then combined with qualitative data obtained from the survey and unstructured feedback. The qualitative data was coded thematically by JSJ using the Consolidated Framework for Implementation Research (CFIR) [19]. Data were coded using an Excel spreadsheet, the CFIR codebook and the domains listed. The qualitative data were placed into relevant domains. Ratings were assigned based on strength and valence of the response. These were then reviewed by the wider research team to reduce researcher bias and improve coding reliability [48].

### 5.6. EMP Procurement and Administration

All EMPs were single-use pre-filled devices. Any one of three antibiotics was used: benzylpenicillin, flucloxacillin or piperacillin/tazobactam. Flucloxacillin and piperacillin/tazobactam EMPs were Vygon Accufuser^®^ (Vygon UK Ltd., Swindon, UK) devices, while benzylpenicillin was supplied in a Baxter LV10 Infusor^®^ (Baxter Healthcare Corporation, Chicago, IL, USA). Various strengths were available from Lloyds Pharmacy Clinical Homecare (LPCH, Coventry, UK).

A simplified version of the procurement process is shown in Figure 4. Pharmacist prescribers from the AMS team ordered EMPs from LPCH on a named patient basis using a pre-populated study prescription. Prescriptions were validated by a pharmacist and processed by the homecare team. Orders needed to be made before 1 pm during the working week (Monday–Friday) to ensure that they would be received within 48 h. No deliveries were made over the weekends. EMPs were delivered to Pinderfields Hospital pharmacy, where they were added to the dispensing system (JAC^®^, Kansas, KC, USA) and sent to quality control/quality assurance (QC/QA) for inspection. Upon release, they were stored in the pharmacy fridge at 2–8 °C until they were required.

An AMS pharmacist prescribed the EMP on Medchart (version 10.1.21.5, Dedalus^®^, Milan, Italy) using a study protocol. EMPs were ordered from Medchart (version 10.1.21.5, Dedalus^®^, Milan, Italy) by pharmacy or ward staff. Prescriptions were validated by another pharmacist and dispensed. Dispensing was completed in the inpatient dispensary, with a maximum of 4 pumps dispensed at a time due to limits on ward fridge capacity. This process was tracked using a prescription tracking system (TMS Insight data capture, Cradley Heath, Dudley, UK). An AMS pharmacy team member then delivered the pumps directly to the ward to ensure they were stored correctly. EMPs were stored in the fridge and removed 2 h prior to administration to allow them to warm to room temperature.

The first EMP administration was observed by the AMS pharmacy team to ensure the SOP was followed and that nursing staff were confident with the training provided. This was planned so it would occur within working hours, Monday–Friday, 9–5. Subsequent administrations were observed for timing purposes where possible.

### 5.7. Training

The AMS team provided a standard operating procedure (SOP) and training to ward nurses, including EMP demonstration and connecting to a PICC line. This was based on current the SOPs used for OPAT.

## 6. Conclusions

This study confirms that it may be feasible to use EMPs to administer IV antibiotics to adult inpatients in secondary care, but further work on this is needed. We have also identified several problems that need to be addressed, including the method for filling the pumps and residual volume. The study has shown that using EMPs in the hospital setting to deliver antibiotics had a high level of acceptability to patients and staff. The study also identified pragmatic factors that supported and challenged the implementation of this intervention. The identification of these factors within other NHS trusts could influence practice developments and highlight the resources needed to introduce EMPs in other hospital settings.

The results from this study provide the foundations for future research, specifically from a health economics perspective, to explore the health benefits of EMPs in relation to cost. It also allows for refinement of the implementation process to ensure future success involving a larger sample of patients.

## Figures and Tables

**Figure 1 antibiotics-14-01122-f001:**
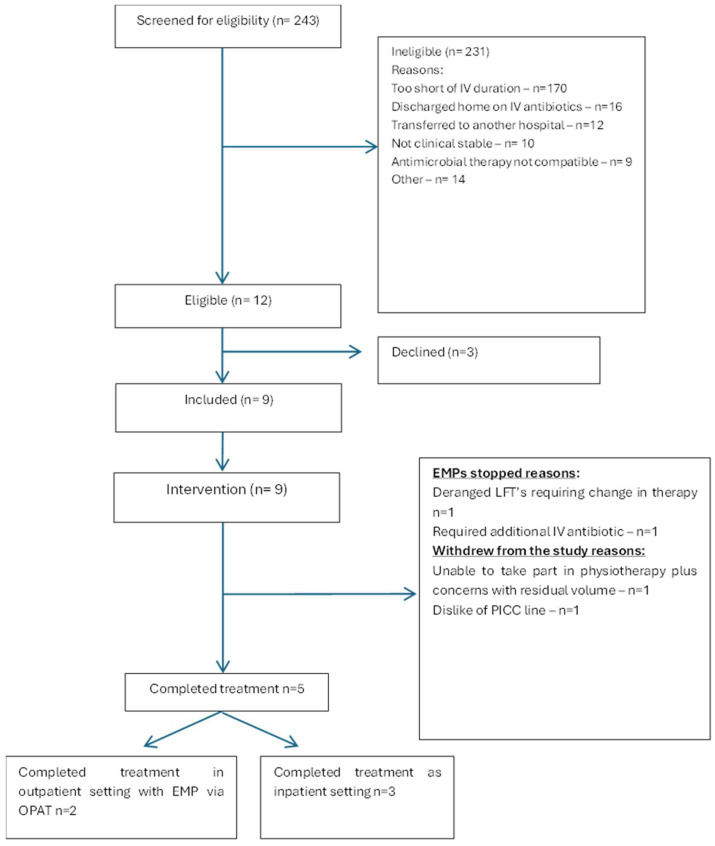
Flow diagram of participants in the study.

**Figure 2 antibiotics-14-01122-f002:**
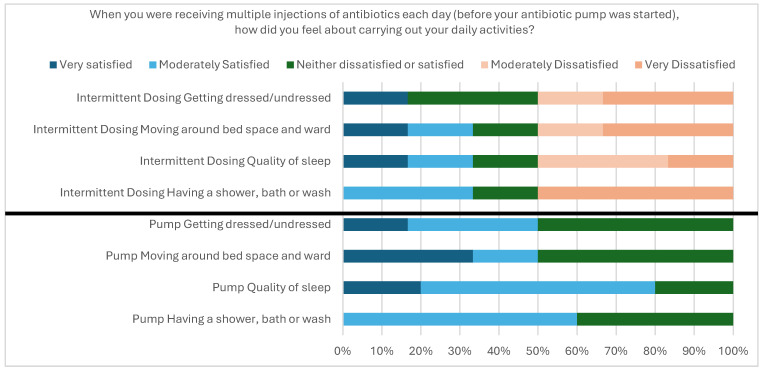
Participant responses to daily activity when receiving multiple (intermittent) dosing or EMP.

**Figure 3 antibiotics-14-01122-f003:**
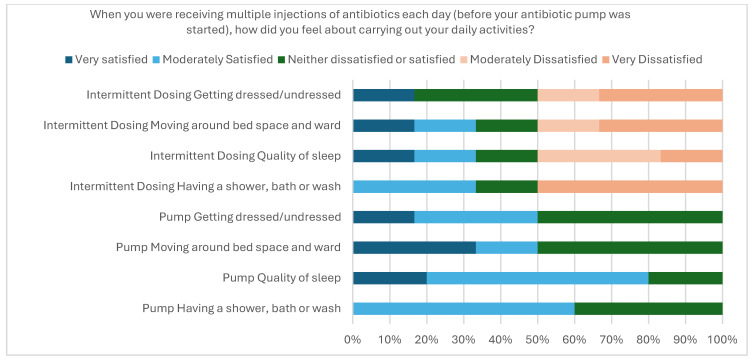
Nurse satisfaction survey of using EMPs for inpatients.

**Figure 4 antibiotics-14-01122-f004:**
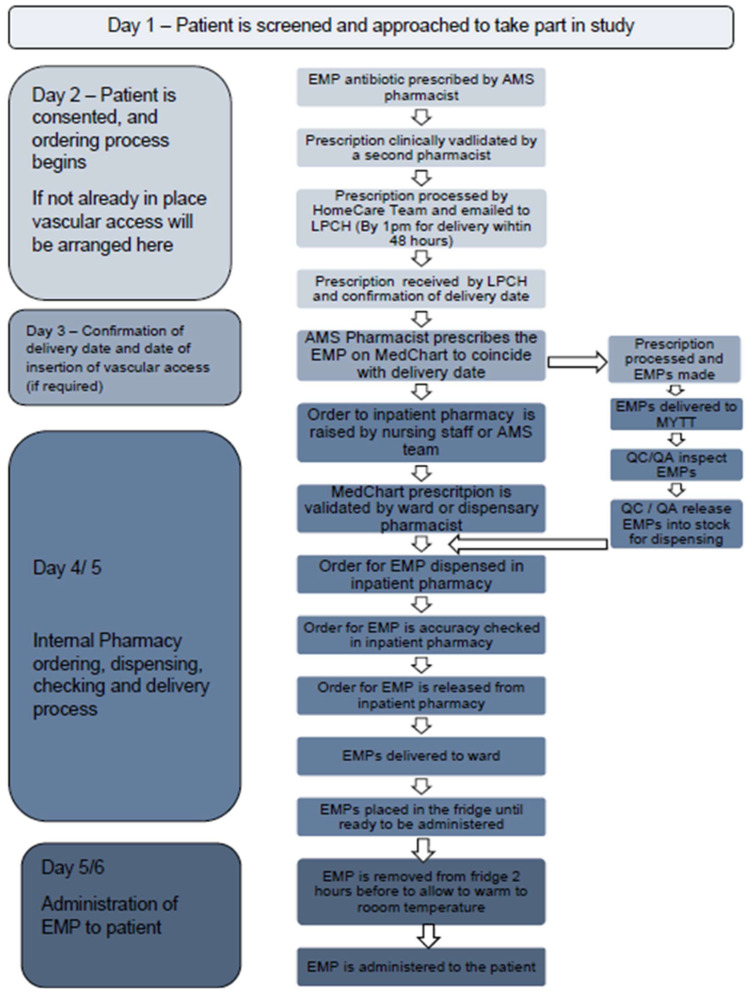
The ordering, dispensing and administration process implemented during the study. The process started once the patient consented, which occurred a minimum of 24 h after being approached.

**Table 1 antibiotics-14-01122-t001:** Study participants baseline characteristics.

Unique ID	Age	Sex	Past Medical History	Weight (kg)	Vascular Access Device	Diagnosis
1	64	M	Aortic stenosis—Tissue aortic valve replacement November 2021, GORD, COPD, two previous hip replacements	84.1	5 French dual lumen PICC line	Endocarditis
2	66	M	Gout, obesity, appendicectomy, PMR, asthma, chronic kidney disease (stage 2)	140	4 French single lumen PICC line	Discitis/infected knee joint
3	67	F	COPD, AF, CCF, type 2 diabetes mellitus, diabetic retinopathy, hypothyroid, diverticular disease, hypoadrenalism, hypertension, GORD	54.8	5 French dual lumen PICC line	Osteomyelitis
4	71	M	Diverticulosis, asthma	61.8	5 French dual lumen PICC line	Discitis
5	59	F	Type 2 diabetes mellitus, previous decompression and fusion surgery at L3/L4, osteoarthritis, hip replacement, recurrent osteomyelitis (right hallux), chronic pancreatitis, IBS, anxiety and depression, alcoholic fatty liver disease, hypertension, hyperlipidemia, diabetic neuropathy, duodenitis	65	3 French single lumen midline	Discitis
6	54	M	Hypertension, type 2 diabetes mellitus, CCF, mitral regurgitation, osteoarthritis, recurrent cellulitis	130	4 French single lumen PICC line	Endocarditis
7	68	M	Diverticulosis, mitral regurgitation, psoriasis, hypertension, type 2 diabetes mellitus	63.8	Single lumen PICC line	Endocarditis
8	63	M	Hypertension, type 2 diabetes mellitus	71.5	4 French single lumen PICC line	Endocarditis
9	78	M	Aortic valve replacement 2012, IHD, AF, pacemaker, type 2 diabetes mellitus	Unable to assess	5 French single lumen PICC line	Endocarditis

**Table 2 antibiotics-14-01122-t002:** Average costs per day of EMP and intermittent dosing.

	Average Costs per Day	Difference Between EMP and Intermittent Dosing Costs per Day (Excluding Delivery Charges)	Difference Between EMP and Intermittent Dosing Costs per Day (with Delivery Charges Included)
EMP (including drug, consumables and nursing costs)	GBP 66.60 (GBP 55.11–GBP 91.45)	GBP 10.45 (GBP −18.71 to GBP 61.38)	GBP 32.50 (GBP 3.35–GBP 83.44)
Intermittent dosing (including drug, consumables and nursing costs)	GBP 56.15 (GBP 30.07–GBP 79.62)		

## Data Availability

The datasets used and/or analyzed during the current study are available from the corresponding author on reasonable request.

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
