# Peer review of "ElastoMeric Infusion Pumps for Hospital AntibioTICs (EMPHATIC): A Feasibility Study"

_antibiotics, 2025, doi:10.3390/antibiotics14111122_

Round 1

Reviewer 1 Report (New Reviewer)

Comments and Suggestions for Authors

Dear Authors of "antibiotics-3903782"

This paper is an interesting article, and I see that a lot of effort went into the design, data acquisition, and presentation. The initial methodology, and therefore the obtained data (retrospective analysis of data by a single institution), is lacking in comparison to an RCT (that may be part of a future effort); however, it does not mean that the conclusion is invalid or unimportant. I believe that an additional paragraph presenting or acknowledging the compatibility between the medications and the elastomeric parts that were used in this study, will address any safety concerns. Since this article presents a novel way of administration, which, unfortunately, leads to a high incidence of residual, uninfused volume of the drug, I believe that an additional, targeted comment should be included. This comment would stress that this is only a feasibility- "proof of concept" study(as this is already written), and in future efforts, outcomes based on additional pharmacological targets of pharmacokinetics and pharmacodynamics ( with indices like Pk, Pd, and comparisons of concentrations in blood with "standard" modes of administration) will be included. I think that this approach will validate your findings even more, especially if clinical treatment failure is considered. A brief comparison between these "low-tech infusion pumps" with proper mobile electronic pumps (like the ones used for PCA infusions) may be incorporated as a comment in this paper and as a standalone future work (even though this differentiation has been addressed in anaesthesiology literature). My opinion is that this article and its concept are very interesting to read, presenting a novel way to disperse antibiotics, but the important drawbacks that are known to anesthesiology (use as PCA pumps) and ICU- Neurosurgery departments (use for baclofen infusion), since the early days of their use, should urge the investigators to strengthen the findings with the use of pk, pd measurements. 

I hope these remarks are found fair, interesting, and worthy of consideration by the authors

Good luck in your endeavors

Author Response

Reviewer 2 Report (New Reviewer)

Comments and Suggestions for Authors

The authors report results of an interesting study, i.e. if elastomeric pumps may be used to administer long course of iv antibiotic treatment and the consequences. While the topic is innovative and interesting, the study clearly lacks from the very small number of patients included, hence this can only be termed as pilot study, and hence the importance of the results as presented are very limited.

The main limitation is that patients only qualified if they required more than 7 days of antibiotic treatment, and even then treatment with EMP was very short. Elastomeric pumps may be safely prepared on the wards if administered immediately, and hence have the authors thought about preparation of elastomeric pumps on the ward by ward nurses? This would safe the nurses still some time, as preparation is only required once, ideally during the first shift, when staffing is usually much better than in the evening. The advantage would be that also shorter courses of antibiotic treatment may be eligible (e.g. 3 days), and cost would probably not increase. The current approach while very interesting may not have any impact if only applicable for a very small fracture of patients (a fracture that might get even smaller if very early iv to po switch is increasing even for endocarditis and osteomyelitis).

Another limitation is the very long time from ordering to receiving a delivery, which is probably the greatest barrier for implementation – again this would not be an issue if the pumps are prepared on the ward. Wasting of pumps would also not be an issue.

Abstract:

  • The conclusion should be amened with the most important limitation – applicability to only a fraction of patients
  • Line 31/32: lack of?

Introduction

  • Short and concise, very good!

Results

  • Table 1: the authors should refer to Appendix A for the pathogens treated and the antibiotic used
  • The data on sleep quality is very interesting – how was this ascertained?
  • Were there any actions taken because of the high residual volume – as this may have come to notice already very early during the study? Did patients wear the Flow Restrictor on the skin at the right place (this is for example important for the Braun Easypump)
  • The same pumps are being used in the outpatient setting – have similar large residual volumes be reported before in the Trust or other countries?

Discussion

  • The limitation section should explicitly state, that due to the very small number of patients included, results are probably only hypothesis generating and have to be treated with caution. This is the most important limitation. Ideally, quality of sleep and other activities of daily living should have been assessed in a random sample of patients that did not convert to EMP treatment (but not in the same patient)
  • The study took place in 2022 – what happened since then? Are inpatient EMPs still used? Have any changes been incorporated?
  • Conclusion: again this should be toned done, as the study does not confirm the feasibility but suggest that it might be possible but further studies with a larger study population are required.

Round 2

Reviewer 2 Report (New Reviewer)

Comments and Suggestions for Authors

The authors have done a great job in revising the manuscript, congratulations, and thank you for taking my concerns seriously. Good luck with your future work.

This manuscript is a resubmission of an earlier submission. The following is a list of the peer review reports and author responses from that submission.

Round 1

Reviewer 1 Report

Comments and Suggestions for Authors

The subject of the study "ElastoMeric infusion Pumps for Hospital AntibioTICs (EM-2 PHATIC) feasibility study" is interesting, but considering the low number of eligible patients, the conclusions are not supported by statistical analysis.

Reviewer 2 Report

Comments and Suggestions for Authors

The manuscript entitled „ElastoMeric infusion Pumps for Hospital AntibioTICs (EM-2 PHATIC) feasibility study” describes a study that investigated elastomeric infusion pumps among inpatients. Topic of this manuscript is an interesting issue, because elastomeric infusion pumps are usually applied to treat outpatients. A major limitation of this study is the small patient number (n=12), however, obtained results are useful.

Comments

1) Selection criteria of patients should be better described.

What is known about underlying diseases of patients in this study? (e.g: diabetes, hypertension)

2) It is mentioned in the text, that flucloxacillin, benzylpenicillin or piperacillin/tazobactam therapy was adminstered to patients, however, were there any other medications applied to patients during the study period?

3) Is there anything known about bacterial pathogens that caused infections in patients of this study?

4)Abstract: „Antibiotic resistance (AMR)”. AMR is the abbreviation of antimicrobial resistance. Please, revise it.

5) References are not required in an abstract. Please, revise this !

6) Abstract: „

 „Patients >18” Do you mean patients >18 years of age ? Please clarify!

7) „OPAT” appears in the abstract, but the full name (phrase) should also appear in the abstract.

8) „Staphylococcus aureus bacteraemia” italic form is needed at the bacterial names: Staphylococcus aureus bacteraemia

Comments on the Quality of English Language

Minor modifications and spell check are needed. However, quality of English language is good.

Reviewer 3 Report

Comments and Suggestions for Authors

The study is well presented, but the number of patient is little. It is possible to have more patients? it is possible to include the period of enlistment (day , months and year)?